# Tracking Progress from Policy Development to Implementation: A Case Study on Adoption of Mandatory Regulation for Nutrition Labelling in Malaysia

**DOI:** 10.3390/nu13020457

**Published:** 2021-01-29

**Authors:** SeeHoe Ng, Bridget Kelly, Heather Yeatman, Boyd Swinburn, Tilakavati Karupaiah

**Affiliations:** 1Early Start, School of Health and Society, University of Wollongong, Wollongong 2522, Australia; shn993@uowmail.edu.au (S.H.N.); bkelly@uow.edu.au (B.K.); hyeatman@uow.edu.au (H.Y.); 2School of Population Health, University of Auckland, Auckland 1072, New Zealand; boyd.swinburn@auckland.ac.nz; 3Dietetics Program, School of Healthcare Sciences, Faculty of Health Sciences, Universiti Kebangsaan Malaysia, Kuala Lumpur 50300, Malaysia; 4School of Biosciences, Faculty of Health and Medical Sciences, Taylor’s University, Subang Jaya 47500, Malaysia

**Keywords:** nutrition, label, policy, food environment, barrier, facilitator

## Abstract

Mandatory nutrition labelling, introduced in Malaysia in 2003, received a “medium implementation” rating from public health experts when previously benchmarked against international best practices by our group. The rating prompted this qualitative case study to explore barriers and facilitators during the policy process. Methods incorporated semi-structured interviews supplemented with cited documents and historical mapping of local and international directions up to 2017. Case participants held senior positions in the Federal government (*n* = 6), food industry (*n =* 3) and civil society representations (*n* = 3). Historical mapping revealed that international directions stimulated policy processes in Malaysia but policy inertia caused implementation gaps. Barriers hindering policy processes included lack of resources, governance complexity, lack of monitoring, technical challenges, policy characteristics linked to costing, lack of sustained efforts in policy advocacy, implementer characteristics and/or industry resistance, including corporate political activities (e.g., lobbying, policy substitution). Facilitators to the policy processes were resource maximization, leadership, stakeholder partnerships or support, policy windows and industry engagement or support. Progressing policy implementation required stronger leadership, resources, inter-ministerial coordination, advocacy partnerships and an accountability monitoring system. This study provides insights for national and global policy entrepreneurs when formulating strategies towards fostering healthy food environments.

## 1. Introduction

Unhealthy diets lead to overweight and obesity and contribute to the development of non-communicable diseases (NCDs) [1,2], which were responsible for 11 million deaths and 255 million disability-adjusted life-years in 2017 [1]. In low- and middle-income countries (LMICs), 85% of premature deaths are from NCDs [3]. Population diets that contribute to the burden of obesity and NCDs, include suboptimal consumption of healthy foods (e.g., fruits and vegetables, nuts and seeds, milk and whole grains) and overconsumption of sugar-sweetened beverages, processed meat, red meat and ultra-processed foods containing high levels of negative nutrients (e.g., sodium, *trans*-fat) [1,4].

Rising sales of ultra-processed foods, such as baked goods and sugar-sweetened beverages, reflect the nutrition transition occurring in Asia, particularly in the South and Southeast Asian regions [5]. High palatability, convenience, extensive marketing and low price of ultra-processed foods contribute to their high consumption [2]. The easy availability, accessibility and desirability of these unhealthy foods drive consumer purchase and consumption behaviors, requiring effective mediation from public sector stakeholders.

Labelling is a policy instrument to guide the food industry to present specific-food product information related to country of origin, environmental protection and consumer health [6]. Some label criteria distinguish high-quality products from uninspected or unaccredited products [6], allowing consumers to make product comparisons. Logically, food labelling policy sets the standard requirements for both locally manufactured and imported products in the market, minimizing issues related to asymmetric product information. However, a theoretical framework analysis suggests that even though a policy may exist, fraudulent behavior of producers will still be observed as regards to improper product information [7]. The existence of such fraudulent behavior likely depends on the enforcement system, public complaints on label violations and economic drivers for food fraud [7].

The World Health Organization (WHO) recommends implementation of mandatory laws and regulations to stipulate nutrition labelling on the back or front of food packages to align with the Codex Alimentarius Commission’s guidelines [8,9]. Previous reviews highlight that most consumers understand and use nutrition labelling during food purchase and selection [10,11], as well as recognize its role in supporting healthier diets [12,13,14]. A nutrition labelling policy governing pre-packaged foods is a cost-effective population intervention, enabling healthy food environments that encourage informed and healthier food choices [12,15]. In recent years, the World Trade Organization (WTO) has been urged to adopt WHO recommendations related to food labelling [2].

Disparities exist in how different countries approach the regulation of food labelling. Of the 124 WHO member countries, 85% have implemented nutrient declarations [16], requiring stating the nutritional content of a food product on the back of the package. However, in practice, policies vary across countries in the nutrients declared and food products for which labels must apply. In Malaysia, the Food Regulations 1985 mandates nutrient declarations only on frequently consumed packaged foods (e.g., bread, breakfast cereals, flour confection, canned products, fruit juices and soft drinks). In addition, foods with special purposes (e.g., infant formula), fortified foods and those carrying nutrition and health claims, as well as ready-to-drink beverages, are mandated to provide nutrient declarations [17]. The list of fully enforced mandatory declarations includes energy, carbohydrate, protein and fat content for frequently consumed packaged foods, together with total sugars content for ready-to-drink beverages and any claimed nutrients [17].

The opinions of local public health experts on the robustness of food environment policies in Malaysia has previously been reported [18]. These experts, using the Food-Environment Policy Index (Food-EPI) tool developed by the International Network for Food and Obesity/Non-communicable Diseases Research, Monitoring and Action Support (INFORMAS), evaluated government implementation of food policies against benchmarks of good practice for healthy food environments [19]. The Food-EPI evaluation rated mandatory nutrition labelling with the highest degree of implementation (61%) but identified implementation gaps [20]. Implementation gaps included the lack of nutrient declarations for added sugar, sodium and saturated fat and that mandatory nutrition labelling only applies to frequently consumed packaged foods. The experts prioritized the need to expand the list of mandatory listed nutrients to include sodium and total sugars, as well as “added sugars” [20].

The Food-EPI evaluation [18,20] highlighted that mandatory nutrition labelling policy in Malaysia was not optimally implemented, leaving room for improvement. This provided an opportunity to investigate the following question: “What are the enabling and limiting factors in the policy process?”, with the view to inform future efforts to shape and progress stronger food labelling policy. This case study aimed to (1) establish for the first time a historical mapping of nutrition labelling policy in parallel with cited international directions up to 2017 and (2) investigate barriers and facilitators in the policy process. Key lessons learnt from this case study on a South-East Asian and an upper-middle-income country will inform national, regional and global policy makers and related stakeholders about positioning strategies for healthy food environments.

## 2. Materials and Methods

A qualitative case study was undertaken, built on semi-structured interviews, together with the review of documents cited by interviewees. Historical mapping of this case against documented international directions up to 2017 guided the findings from the semi-structured interviews. This study received ethics approvals from the Research Ethics Committee, The National University of Malaysia (UKM PP1/111/8/JEP-2016-394); the Social Science Human Research Ethics Committee of the University of Wollongong (HE16/297); and the Medical Research and Ethics Committee, Ministry of Health Malaysia (NMRR-17-195-34142(IIR)). Consenting participants gave signed informed consent and researchers adhered to the anonymity of their identity.

### 2.1. Stages of Execution

#### 2.1.1. Stage 1: Theoretical Basis of the Interview Guide

The *Advocacy Coalition Framework* [21] informed the development of the semi-structured interview guide. This approach recognizes the potential interplay of three stakeholders, the food industry, government and civil society, in food environment subsystems [19]. The framework includes consideration of coalition members’ core beliefs, resources, coalition inter-relationships, relative stable parameters for policy shaping and external events affecting policy processes. Posing open-ended questions such as ‘Who was involved?’ and ‘Was there any key event which might have precipitated it to happen?’ elicited relevant information.

Additionally, the interview guide incorporated aspects of the *Model of Agenda Building* [22]. An example of an open-ended question was ‘What can you tell me about the process?’. This facilitated understanding about the policy’s initiation and added an interpretative dimension to the coalitions’ beliefs and resources. Finally, the interview guide included elements of the *Theory of Coalition Structuring*. For instance, an open-ended question of ‘What were the key arguments and supports…?’ offered a window to explore the coalitions’ internal structures. It provided insights to facilitate the interpretation of *transactions* (cost-benefits analysis), *relationships* (motivating factors and basis of cooperation) and *controls* (resources management capacity i.e., members and its opponents) of the coalitions [23,24].

Integration of these theoretical frameworks to form the interview guide was beneficial to catalyze comprehension of the investigated policy, explain the evidence related to the past efforts, review the current status quo and inform future actions. The interview guide was designed to collect data for two case studies (see Appendix A). The findings related to mandatory nutrition labelling are presented here.

#### 2.1.2. Stage 2: Data Collection

*Sourcing data*—Direct requests were sent in July 2017 to leading government agencies responsible for labelling policies. Free document access to official government documents was restricted by the Malaysian *Official Secrets Act 1972* [25]. Therefore, preliminary historical mapping used publicly available information, later verified and amended during the government stakeholder consultations and interviews.

*Setting up the interview and participant characteristics*—Semi-structured interviews were performed by one researcher (SHN) through face-to-face sessions between June 2018 and February 2019. Government agencies relevant to the case nominated potential participants. Later, engaging the snowball sampling, potential new participants were nominated by the first tier of participants. Nominated participants comprised representations from government, industry and civil society (inclusive of academia, professional bodies and non-government organizations).

The selection criteria required participants to possess at least five years of work experience related to the policy area, declare conflicts of interest and permit the interview to be audio-recorded. In total, nineteen potential participants were identified through saturation of the snowball sampling method (e.g., the same name being nominated repeatedly). All potential participants were contacted through official invitation letters that included information sheets and study brochures. Follow-up occurred through emails and phone calls. Four invitees declined participation for reasons including poor health, non-regulatory background, retired or on leave, whilst three invitees did not respond.

*Interview process*—Participants first filled in biographical details, followed by screening questions to determine their appropriateness for the interview. The interview started with memory mapping [26], which involved a chronological presentation of historical mapping of mandatory nutrition labelling to initiate a stimulus for recall. The interview utilized oral history techniques to facilitate the flow of discussion [27].

*Corporate political activities and other probings—*Upon completing replies to questions in the discussion guide, only non-industry participants were further probed regarding their perceptions on corporate political activities of the food industry, as recommended by Mialon et al. [28] The final questions addressed all participants on the importance of monitoring, recommendations for potential key informants and relevant publicly available materials.

All interviews were conducted in English and audio-recorded with written notes to support the recording. The interview was voluntary and did not involve any monetary payment to participants.

#### 2.1.3. Stage 3: Data Transcription, Consolidation and Analysis

Audio-records were transcribed verbatim (SHN) and cross-checked by another researcher (TK) for logical consistency. Only nine participants elected to verify the transcripts, of which six participants made amendments after verification to improve clarity or censor statements to protect anonymity. All transcripts were managed and analyzed using the qualitative data analysis software NVivo 12 (QSR International, Australia 2018). Transcripts were coded thematically using the constant comparison analysis approach described by Leech and Onwuegbuzie [29]. The approach involves the development of guidelines to build “free nodes” of data, not been previously assigned, and which are then grouped to build the “tree node” using the NVivo software. Barriers and facilitators in the implementation of food environment policies identified through a systematic review undertaken by the research team [30] also informed the development of themes.

The literature search extended to publicly available documents. Retrieving documents from websites, archives, guidelines and legislation were recommended by participants during the interview. The literature built up thus incorporated information from international agencies (e.g., WHO, Codex and *Consumer International* documents) and national documents (e.g., government publications, memorandum, bulletins, newspapers and web pages).

SHN prepared the preliminary findings, which were then reviewed by BK, HY and TK for data saturation, credibility and dependability of interpretations based on their expertise and/or policy experience. A subsequent step verified the preliminary results with relevant government agencies. Based on feedback, no major revisions were required. Minor amendments to improve clarity were made to the historical mapping.

## 3. Results

The case study findings are presented in two parts. Part I explains the historical mapping for this case against international directions. Part II explores the thematic findings related to policy process for this case, followed by a summary of five recommendations for stakeholders to progress future food labelling policy processes.

### 3.1. Participant Characteristics

Twelve people participated, representing the Federal government (*n* = 6), food industry (*n* = 3) and civil society (*n* = 3). Participants had a mean age of 54.7 ± 11.1 years and 24.9 ± 11.0 years of experience in the related field (see Appendix A). All participants had completed tertiary education, attaining Bachelor’s (*n* = 4), Master’s (*n* = 5) and Doctoral (*n* = 3) degrees. Food regulations, policy development, nutrition and public health were common areas of expertise of the participants. As well as providing information on the food labelling case, nine participants also gave evidence on corporate political activities. Interviews lasted an average length of 1 h 14 min.

### 3.2. Part I: Historical Mapping of Mandatory Nutrition Labelling Case

Participants reported three key international directions occurring between 1985 and 1995. These included: (a) *Codex Guidelines on Nutrition Labelling*, promoting declarations on energy, carbohydrate, protein and fat content (termed as the ‘Big 4’) in 1985 [31]; (b) the *World Declaration on Nutrition 1992* urging countries to harmonize with Codex food label requirements [32]; and (c) the establishment of the WTO in 1995, which endorsed Codex guidelines [33]. Case participants noted a lack of significant response in Malaysia to these events during this period, although they did stimulate some policy discussion. Participants commented on the rising trends of overnutrition and NCDs from the 1980s to the 1990s [34,35,36]. Some participants identified government-led prevention actions during this period, such as *Healthy Lifestyle Awareness* campaigns (1991–2002) that included education on reading food labels [37,38].

Some participants identified government actions during the period following the release of the international nutrition directives. They cited the *National Plan of Action for Nutrition of Malaysia* (NPANM) 1996–2000, which recommended to align with the Codex requirements for nutrition labelling [35]. In about the year 2000, the government proposed an amendment to the *Food Regulations 1985*, followed by multiple engagements with relevant stakeholders [39,40,41,42]. In 2003, mandatory nutrition labelling (P.U.(A)88, Reg.18B) was gazetted to extend the enforcement date to June 2005 [43,44]. The *Guide to Nutrition Labelling and Claims* was first published in 2005 [45] and updated twice [17,46] to facilitate policy implementation in Malaysia.

On the international front, the *WHO’s Global Strategy on Diet, Physical Activity and Health* (DPAS) [47] triggered a revision of the Codex guidelines. The Codex guidelines in 2011 expanded mandatory nutrient declarations to include total sugars, sodium and saturated fat [48,49,50]. In Malaysia, a review of legislation for the list of mandatory foods and nutrients was planned under the NPANM II 2006–2015 [51], in tandem with a strategy under the *National Nutrition Policy of Malaysia* [52]. Between 2008 and 2009, Malaysian action was dithered on the *trans*-fat labelling proposal [53] but gazetting the “fatty acids” formatting and “total sugars” definition [54]. The Cabinet approved the mandatory nutrition labelling which extended to include instant noodles [53].

Participants considered that some of the international directives post 2010 were important. For instance, governments globally became committed to the position to enable informed food choices via nutrition information in the *Rome Declaration on Nutrition 2014* [55]. In terms of advocacy, Consumers International urged the establishment of a global convention for healthy diet and alignment of Codex principles. They also conducted a campaign with a “healthy diet” theme [56,57]. In Malaysia, NPANM III 2016–2025 set plans for the future introduction of mandatory sodium and total sugars declarations for all food products and declaration of four types of fatty acids relevant to four food categories [58]. While these plans were in gestation, the introduction of a voluntary *Healthier Choice Logo* (HCL) in 2017 required endorsed products to display the relevant nutrient declarations as per HCL criteria [59,60].

Figure 1 provides a historical mapping of the events discussed for the mandatory nutrition labelling case. For detailed description of this case, please see Appendix A.

.

### 3.3. Part II: Thematic Analysis of Case

Policy processes comprise the stages of policy development, implementation and/or future plans. Overall, seven themes emerged in this case relating to policy processes, including (i) policy commitment, (ii) policy governance, (iii) external policy organization, (iv) society, (v) industry, (vi) policy specific issues and (vii) opportunistic advantages. Part II describes the barriers and facilitators to (a) policy development and (b) implementation and/or future plan periods (Table 1).

#### 3.3.1. Policy Commitment

“Lack of resources”, the nature of “implementer characteristics”, and “lack of sustained effort”, were three barrier sub-themes that emerged in relation to policy commitment. “Resource availability or maximization”, “supportive organizational action” and “leadership” were facilitator sub-themes.

Participants reported “lack of resources” such as technical knowledge and expertise, local evidence, guidelines, funding due to low prioritization for monitoring and evaluation and/or laboratory capacity, which restricted policy processes.

“Implementer characteristics” were linked to industries’ capacity and readiness to accept nutrition labelling hindered the policy process. Specific to policy implementation, competing priorities experienced by the enforcement team was another barrier linked to “implementer characteristic”. These issues were reflected in the following opinions:

“They (SME) do not understand (the new regulations)… (said) very difficult… (when) discussion, never came… do not have QC (Quality Control)… (and) wait until big companies to implement (first)…” (Civil society and industry, Development and Implementation).

“(Based on) the hierarchy (of duties) … more enforcements related to the food safety… (rather than) nutrition labelling…”, (Government, Implementation).

A “lack of sustained effort” by NGO advocacy during policy implementation was reported. Participants repeatedly highlighted concerned agencies lacked uniformity in messaging consumer education, as well as sustainability issues.

In terms of facilitators, participants recognized that “resource availability or maximization” was critical throughout the policy processes. For instance, Codex guidelines, WHO recommendations and/or other ASEAN country experiences benefited stakeholders by facilitating the policy processes in Malaysia.

“Supportive organizational action” from top management and presence of structured systems modelled on the Codex Committees facilitated the policy process. Specific to policy development, “leadership” from the government was cited as a facilitator sub-theme for mandatory nutrition labelling. A participant highlighted that:

“Last time, (nutrition labelling) was not mandatory… (but) Malaysia was the first country in ASEAN (Association of Southeast Asian Nations) to have mandatory … We are bold enough to do that.” (Civil society, Development).

#### 3.3.2. Policy Governance

“Complexity” and “lack of monitoring” were two barrier sub-themes identified within policy governance specific to policy implementation. “Strategies in policy process” was the only facilitator sub-theme identified within policy governance.

“Complexity” arises from issues in integrating labelling policy into existing regulatory frameworks that mainly mandated food safety. Participants cited complex bureaucratic procedures for legislation and competition with other policies as highlighted by this opinion:

“I think there are internal issues, could be bureaucratic… political… legal. The Attorney General may come back and say, “You need to tackle (these), frame those words”. Then, they have to take (the policy) back again and take actions.” (Civil society, Implementation).

Participants also acknowledged “lack of monitoring” as another barrier during policy implementation. This was likely to be one of the contributing factors to policy inertia.

“Strategies in the policy process” was a facilitator sub-theme, which included media dissemination, development of guidebooks and permitting industry to negotiate for flexible grace periods for policy enforcement. Even though the Codex guidelines informed policy development and aligned with trading purposes, the Guidelines were adapted to the local context of Malaysia. Comments reflecting these views were:

“It is not 100% we adopted the Codex Guidelines… Codex is the reference… (is useful) if there is a dispute at the WTO (World Trade Organization)… we have (been) sensitizing… we had the roadshows… discussion with the importers… called all the embassies to inform them.” (Government, Development).

“Two years grace period… the educational enforcement… in our guideline, even though you do not send for the lab analysis, you can use the food composition (database calculation method for labelling)… (and the guideline set) the analytical tolerance…” (Government and industry, Implementation).

#### 3.3.3. External to Policy Organization

“Stakeholder partnership or support” was the only facilitator sub-theme identified with this theme. This facilitator sub-theme was observed at the intra- and inter-ministerial levels (e.g., research institutions, trade and consumerism related agencies), as well as involving civil society members to advocate for the Big 4 declarations during the policy process. In addition, participants highlighted the role of the International Life Sciences Institute (ILSI), a non-profit organization with members primarily from industry. A participant described ILSI’s role during policy process as:

“ILSI decided, because this (Seminar) was regional… ILSI was willing to offer a platform, to bring in the various stakeholders…what’s news in this are … (bring) awareness and education to the professionals… (inviting) overseas speakers and (local government stakeholders).” (Government and industry, Development and Implementation).

#### 3.3.4. Society

Two sub-themes were found within the society theme. “Low demand or other attributes” were identified as a barrier sub-theme, whilst the facilitator sub-theme was “social acceptance, awareness or benefit”.

Participants frequently mentioned “low demand or other attributes” related to consumer understanding and their limited use of nutrition labels, which they perceived to hinder policy processes. The following opinion reflected these views:

“The level of understanding in our consumers (was) different from the Western… at that time (of policy development)… (nutrition) labelling was very new… (and focused more on) underweight… maybe they (consumers) read the label (now), but to what extent (do) they read the label?” (Civil society and government, Development and Implementation).

In contrast, “social acceptance, awareness or benefit” underpinned the development of nutrition labelling, particularly its positive effects linked to consumer education and healthy diets.

#### 3.3.5. Industry

Two opposing sub-themes were reported within the industry theme. “Industry resistance” as the barrier sub-theme, whereas “industry engagement or support” was the facilitator sub-theme.

“Industry resistance” was observed during the initial policy development step, contributing to limiting the focus to just the Big 4 declarations before broadening to include other nutrients of concern. Even after gazetting the mandatory nutrition labelling in March 2003, “industry resistance” included the raising of issues such as waste from old packaging, which resulted in the extension of full enforcement of food regulations to June 2005. Nine non-industry participants remarked on the occurrence of industries’ corporate political activities. Strategies such as policy substitution and lobbying were cited by some participants to influence policy implementation, as reflected by the following views:

Corporate political activity—policy substitution: “… related to nutrition declaration… they (industries) did not agree (with) the level… they proposed to have it in the Guideline, not in the regulations.” (Government, Implementation).

Corporate political activity—information and messaging (lobbying): “They (industries) are represented as a group… not by individual companies… Well, you can say they lobby…” (Civil society, Implementation).

Participants with a civil society background expressed contrary views with regard to “lobbying”, with some identifying it as a positive contribution towards policy development, while others considered it as industry interference and manipulation of policy outcomes. These opposing views were reflected by the following comments:

“(As they are) speaking on front voices for the industry… we should not necessarily use the word “lobbying” in the negative sense…”


*versus*


“many lobbying… obviously not (only in) Malaysia but worldwide… powerful lobbying to make sure (policy) does not work… even developed countries… have to deal with this food lobbying, which is extremely powerful and very well coordinated.”

Participants identified “industry engagement or support” as a facilitator during the policy process because of the likely low negative impacts of limited nutrition labelling. Favorable opinions were the need for standardization for a fair-trade environment, marketing opportunities with revenue benefits to industry on a long-term basis, availability of platforms for industry engagement and industry readiness factors, in terms of the ability to perform proximate analysis and/or have established products align to labelling standards. With regard to implementation and/or future plans, industry participants in this study showed positive views on the mandatory nutrient list expansion. Possible reasons cited were:

“Because most of our customers (e.g., foreign retailers) required us to provide based on the importing country’s regulation, (which the) requirement is higher (than) the Malaysian (standards).” (Industry, Implementation and/or future plan).

“(Nutrition labelling) is part of the cost of goods, but it is one-off… It is not very expensive to analyze sodium and total sugars (contents)… Printing is routine… If (the) right transition period (is provided), there is not really a big issue because most of the companies are changing their packaging.” (Industry, Implementation and/or future plan).

#### 3.3.6. Policy Specific Issue

Under the policy-specific issue theme, there were two sub-themes specific to barriers. These included “policy characteristics” and “technical challenges”. No facilitator sub-theme was identified.

“Policy characteristics” linked to costing (e.g., analytical, labor and printing costs), uncertainties in specifications like labelling format and implementation timeline, as well as slow regulatory amendment procedures which hindered the policy process. Comments reflecting these views were:

“To (standardize the) label, of course there will be some costing. Because (industries) have to send to lab for analysis… change labels…” (Government, Development).

“The analytical declaration tolerance, it (was) only after implementation, we (industry) realized it (as a problem). There was quite a bit of discussion.” (Industry, Implementation).

“The amendment was quite slow… we pushed for sodium and also sugars (labelling) for quite some time.” (Government, Implementation and/or future plan).

“Technical challenges” hindered the policy processes. These included a lack of laboratory capacity, analytical limitations, a non-comprehensive food composition database for nutrients of concern, stock turnover issues and/or challenges related to standards’ harmonization.

#### 3.3.7. Opportunistic Advantages

“Policy windows” and “revenue-related effects” emerged as two sub-themes for opportunistic advantages, both of which were discussed as facilitators.

Rising obesity and NCD rates triggered the policy process. The international directions (e.g., Codex or WHO related opportunities), coupled with local events such as unregulated claims, the appointment of Food Safety Quality Division as the Codex contact point and lobbying by international academia for salt reformulation contributed to facilitating the policy process. All of these factors formed “policy windows” for mandatory nutrition labelling in Malaysia.

“Revenue related effects” facilitated the policy development of mandatory nutrition labelling. Participants’ comments were influenced by their backgrounds, such as:

“Driving force… Of course, it is facilitating the trade as well.” (Government, Development).

“Companies feel… “If I do not have it, I lose out to the other companies.” (Civil society, Development).

#### 3.3.8. Recommendations

The interviews also generated recommendations from participants to facilitate future food labelling policy processes (Table 2). Actions recommended by participants included enhancing consumer education, determining an appropriate transition period for full enforcement, maximizing resources particularly scientific evidence, publishing clear guidelines with stakeholders’ engagement and introducing proper training, and to intensify accountability systems.

## 4. Discussion

This case study provides an in-depth understanding on the policy processes leading to mandatory nutrition labelling in Malaysia. Uniquely, this study applied an integrated theoretical framework that overcame the individual limits of the single theory, model, or framework and offered a convergent analytical overview of the policy processes. The novel use of historical mapping in case analysis tracked parallel interactions between local events of mandatory nutrition labelling and international directions. Later, drawing on the lived experiences of multiple policy stakeholders, this case study further lends important insights into the barriers and facilitators of policy processes likely occurring in LMICs such as in Malaysia. The integrated theoretical framework situated the interview data concurrent to the timeline of significant events and publicly available information, revealing critical information to facilitate a better understanding of the local mandatory nutrition labelling policy processes.

Participants identified more facilitators (*n* = 9) and fewer barriers (*n* = 6) during the development of mandatory nutrition labelling policy, whereas they observed a reverse trend during policy implementation and/or future plans. Governance “complexity”, “lack of monitoring” and “lack of sustained efforts” in consumer education and policy advocacy were emerging barriers specific to policy implementation. Worth noting, two of these former barriers were also identified as the most cited barriers in LMICs, whilst implementing food environment policies [30,61]. The issues challenging the policy processes found in this study, namely SME capabilities, costs, laboratory capacity and product turnover, also have been reported by others [62,63,64,65]. Poor consumer understanding of nutrition labelling in Malaysia may diminish community demand for food labelling. However, participants commented that consumer education over a long period would overcome this barrier.

The government’s early development of nutrition labelling policy clearly reflected and followed the international directives. Malaysia fulfilled the international commitments [32,33] and pioneered mandatory nutrition labelling amongst the South-east Asian countries, harmonizing this with *Codex Guidelines on Nutrition Labelling 1985* [65,66]. The enactment of the Big 4 declarations in 2003 was the initial step to strengthen nutrition labelling in the marketplace. In contrast, more developed countries such as Australia and New Zealand mandated additional nutrient declarations (i.e., sodium, sugars and saturated fat) as part of their public health prevention efforts [67] during the same period. The question is—why was nutrition labelling only limited to the Big 4 declarations? This may reflect local constraints in Malaysia but could also be attributed to Codex guidelines setting minimum requirements to enable less-developed nations achieve primary food safety as part of provisions to facilitate trade [68]. However, the limited nutrient declarations may also reflect constraints from political alliances and the presence of the food industry front group represented by ILSI during Codex meetings, as has been reported previously [68]. Thow et al. [69] shared concerns on the high representation of food industry compared to public health advocates in Codex meetings and called for the latter to raise awareness of industry influence over domestic policymakers.

Although the NPANM III 2016–2025 [58] provided the impetus for significant policy activities to occur, the policies still took a relatively long time to progress. The Malaysian progress encountered policy inertia from the initial step of the Big 4 declarations in 2003, despite setting significant plans for legislation in 2016 [58]. In a recent amendment to the *Food Regulations 1985*, the list of mandatory nutrients expanded to include total sugars and sodium declarations and covered more food categories, with a likely full implementation date to be in July 2022 [70]. Even though WHO recommendations and Codex guidelines were acknowledged to facilitate the policy process in Malaysia, the scaled-up action process took nearly 18 years, implying policy inertia. The slowness in policy implementation was attributed to ground-level challenges, including low priority for resource use, complex bureaucratic procedures, unsustainable advocacy efforts, industry’s capacity and/or resistance from industry. In contrast, for a similar policy experience in Canada, the period between policy formulation to shift from voluntary to mandatory nutrition labelling policy and the full implementation of the new mandatory nutrition labelling regulation took ~8 years, despite facing opposition from industry [71].

“Industry resistance” is a commonly identified barrier encountered during the policy processes of food environment policies in South-East Asian countries including Thailand [72] and the Philippines [73]. This barrier was also evident during the policy processes for mandatory nutrition labelling in Malaysia. Participants discussed corporate political activities such as lobbying and policy substitution as techniques to “buy time” and/or “water-down” the regulations. An observation in this study was that participants were divided on lobbying by industry, some indicating industry actions were supportive, while others expressed concern about their influence. Ronit and Jensen [74] explain that industries could easily act to prevent the enactment of more binding public regulation, to mitigate any possibilities a policy would result in a higher cost or lesser profit to them. In the recent Food-EPI evaluation for Malaysia, public health experts were rated “low” on the implementation of national governance oversight on commercial influences [18]. Other research in Malaysia also found that few food industries were committed to practicing corporate social responsibility activities without branding and product promotion, and abstain from making political donations [75]. Thus, such concerns of industry interference are not baseless. Policymakers need to be cautious about the power of lobbying, which contributes to policy inertia.

Participants, irrespective of government or academic background, recognized the positive role of the industry-funded ILSI in disseminating information on food labelling updates. However, recent studies have identified the high risk that ILSI poses to compromising public health outcomes, due to this body’s very strong ties with the food industry [68,76,77,78]. Commercial interests influence health by truncating policies centered on public interests and might potentially result in policy inertia. Tempels et al. [79] suggest that public–private partnerships for health should factor in ethical reflections in terms of conflicts of interest and encourage a wider debate on corporate social responsibility actions in public health issues. Such deliberations were not reported in this case study. A conservative mechanism based on the principles of WHO [80] and Cullerton et al. [81] may be necessary to manage conflicts of interest in public–private partnerships.

Seeking participants’ recommendations based on their policy experiences provides valuable guidance for future food policy actions. Participants emphasized the need for independent monitoring as well as maximizing resources, especially credible evidence (e.g., Codex guidelines, WHO recommendations and local research) to progress future food labelling policy processes. Literature from the same region [72,73] supports similar views on the importance of resources to inform policy decisions, alongside a monitoring and evaluation mechanism to identify the gaps in food environment policy implementation. In addition, a majority of the participants recommended strengthening consumer education activities in promoting awareness and use of nutrition labelling. A systematic review of nutrition label education studies in Western countries supports this recommendation and found that consumer education could positively impact a consumer’s label understanding and its use [82]. Over time, these impacts are anticipated to establish impetus for the government and food industry to change the status quo, fulfilling the social and market demands for comprehensive nutrition labelling.

Implementation of the mandatory nutrition labelling policy still warrants further improvement in Malaysia, despite progressive policy steps over the last few decades. This was in spite of past experiences related to the labelling area, increasing market demands for comprehensive labelling and establishing optimal laboratory capacity, either in-house or outsourced to accredited laboratories. For instance, nearly half of prominent food companies (13 out of 28) had declared total sugar content on some products [75], well before full enforcement of the new regulation amendments in 2022 [70]. For food industries yet to implement the policy, participants recommended that stock turnover issues could be tackled with an appropriate grace period. Drawing from past Malaysian experience, a maximum of a 2-year transition period to enforce total sugars and sodium declarations as per the recent gazette [70] would enable businesses to phase out existing label inventory and align newer labelling with mandatory requirements. Such a grace period duration is in line with the recommendation of the *Codex Committee on Food Labelling* [63] for governments. Participants also recommended the need for clear guidelines and industry engagement and training, particularly to support SME businesses. This study’s recommendations are in line with the Codex views [63] towards creating effective communication strategies to allow synergistic effects for progressing policy implementation.

Overall, this study provides a timely assessment to document and analyze critical experiences of key informants, as witnesses and stakeholders, involved over the past two decades in mandatory nutrition labelling in Malaysia. Findings from this study are potentially generalizable to other food policy areas, and to other Asian countries with similar economies but in a local context.

A study limitation was the inaccessibility to government documents, which restricted accuracy of information to only historical mapping of evidence in the policy processes. To overcome this limitation, the study applied an integrated theoretical framework to develop the discussion guide and probes for important points during semi-structured interviews. Key points such as local and external events, resources and a basis of cooperation between stakeholders were explored using the framework during the semi-structured interviews. These data were coupled with publicly available information to assist the mapping arrangement and verification of preliminary results with concerned government agencies.

The small sample size of 12 interviewed participants, may also limit data interpretation and extrapolation. Our sampling was limited to the small number of individuals involved in or with knowledge about food policy processes. However, a small sample size is inherent to case study interviews related to food environment policies [68,83,84,85,86]. Despite this limitation, recruitment ensured adequate representation for seniority and diverse backgrounds related to government, industry and civil society.

## 5. Conclusions

This study adds insights into the barriers and facilitators in the mandatory nutrition labelling policy processes from an LMIC perspective. The case study revealed the main influences on the policy processes to be policy commitment, governance and its technical and specificity issues, stakeholders’ relationships, social attributes and impacts, food industry’s policy position, as well as opportunities linked to local and external triggers that influenced policy processes. Policy inertia was evident in this Malaysian experience. Key lessons gained from this study can inform policy entrepreneurs, particularly in LMICs, to understand considerations of adopting mandatory nutrition labelling and formulate strategies to mitigate challenges and seize opportunities to create healthy food environments. Future research directions are necessary to examine the impact of nutrition labelling policies on reformulation, sales and consumer behaviors using quantitative analyses, evaluate corporate political activities of food companies and peak bodies, as well as related influences on policy inertia and mechanisms to manage conflicts of interest.

## Figures and Tables

**Figure 1 nutrients-13-00457-f001:**
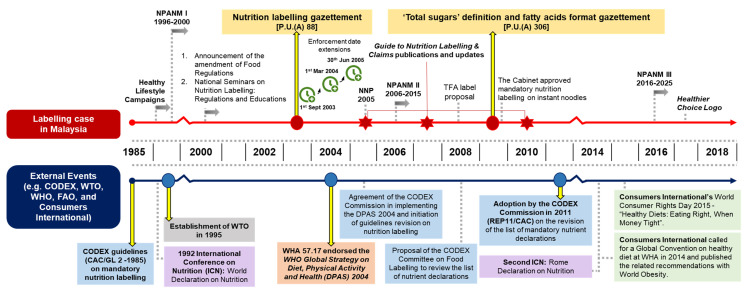
Historical mapping of mandatory nutrition labelling in Malaysia Abbreviations: DPAS = WHO Global Strategy on Diet, Physical Activity and Health; FAO = Food and Agriculture Organization; ICN = International Conference on Nutrition; NNP = National Nutrition Policy of Malaysia; NPANM = National Plan of Action for Nutrition of Malaysia; TFA = trans fatty acids; WHA = World Health Assembly; WHO = World Health Organization; WTO = World Trade Organization.

**Table 1 nutrients-13-00457-t001:** Summary of barriers and facilitators as per stage of policy process.

Theme	Sub-Theme	Policy Process
Development	Implementation/Future Plans
Policy commitment	☹	Lack of resources	√	√
☹	Implementer characteristics	√	√
☹	Lack of sustained efforts	X	√
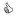	Resource availability or maximization	√	√
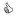	Supportive organizational action	√	√
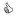	Leadership	√	X
Policy governance	☹	Complexity	X	√
☹	Lack of monitoring	X	√
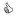	Strategies in policy process	√	√
External to policy organization	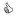	Stakeholder partnership or support	√	√
Society	☹	Low demand or other attributes	√	√
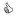	Social acceptance, awareness or benefit	√	X
Industry	☹	Industry resistance	√	√
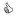	Industry engagement or support	√	√
Policy specific issue	☹	Policy characteristics	√	√
☹	Technical challenges	√	√
Opportunistic advantages	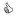	Policy window	√	√
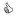	Revenue related effects	√	X

Symbols: ☹ = barrier; 
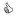
 = facilitator; √ = identified; X = not identified.

**Table 2 nutrients-13-00457-t002:** Recommendations for stakeholders to progress future food labelling policy processes.

Recommendations	
•	Build a cohesive effort in consumer education by involving both non-nutrition and health related agencies in mass education and awareness of nutrition labelling with SMART targets.
•	Set an appropriate transition period to change new packaging aligning with minimum order quantities of the food industry (e.g., a grace period ranges from 6 months to 2 years).
•	Align resource allocations with credible international norms and focus on representative local scientific evidence, including pre- and post-policy implementation surveys on consumers’ feedback on nutrition labelling, to inform policy decisions.
•	Provide updates and clear guidelines, as well as enhancing stakeholders’ engagement and training to support policy adoption, in particular the SME businesses.
•	Strengthen monitoring and enforcement systems to hold the food industry to account for providing accurate nutrition labelling to consumers in making healthier food choices.

Abbreviations: SMART = Specific, measurable, achievable, relevant and time bound; SME = Small and medium-sized enterprises.

## Data Availability

Not applicable.

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
