# Peer review of "Tracking Progress from Policy Development to Implementation: A Case Study on Adoption of Mandatory Regulation for Nutrition Labelling in Malaysia"

_nutrients, 2021, doi:10.3390/nu13020457_

Round 1
Reviewer 1 Report
This study qualitatively assessed barriers and facilitators of mandatory nutrition labelling policy processes in Malaysia, and mapped the history of nutrition labelling initiatives internationally and in Malaysia. This is an important topic with implications for policy, and is appropriate for publication in Nutrients based on qualitative nutrition labeling analyses that have been published in the journal previously. I have included notes below on areas of the paper that I believe could be strengthened:
ABSTRACT
Can you provide more context in the first sentence of the abstract explaining when mandatory nutrition labeling in Malaysia was implemented, and what “medium implementation” means? Could you also explain who conducted the recent food environment policy analysis?
INTRODUCTION
p.2, lines 34-40: You say that population diets are “perceived as unhealthy”—who perceives them as unhealthy? Do you mean to say that those diets lead to NCDs?
p.2, line 47: Can you provide an explanation of what you mean by “nutrition labelling” here? Is this a nutrition facts panel on the back of the package? Or front-of-package labeling? Similarly, in the next paragraph, can you provide more explanation about what “nutrient declarations” are?
p.2, lines 55-62: Which nutrients are required to be listed under Malaysian regulations?
p.2, line 70: What does “limited application” mean here?
p.2, lines 72-79: Did the INFORMAS report ultimately conclude that the policy was not properly implemented? Or is the takeaway that the policy itself should just be stronger? In other words, was the goal of your study to understand why the policy was not implemented completely, or why the policy didn’t include added sugar disclosure, etc? Can you provide more context for why historical mapping of nutrition labelling policy was one of your goals? Has that not been documented previously?
METHODS
p.4, 2.1.3: If you interviewed 19-7 people (12), why were only 9 transcripts verified by interviewees?
RESULTS
p.4, 3.2. Part 1: What are “Big 4 declarations in 1985”? Can you provide more explanation about what Codex standards are? Are these general international actions on nutrition labeling unrelated to Malaysia, or are any of them specific to the country?
p.7, Table 1: Could you restructure Table 1 so that sub-themes are grouped by barriers and facilitators?
pp.7-11: It is difficult to read the direct text from all participants—could you summarize the subthemes instead of just listing all of the quotations, and only feature particularly salient quotations?
DISCUSSION
Much of the discussion summarizing the overall results could be moved to the results section (see my previous comment), and then the discussion could focus more on interpretation and comparing results to other studies.
Author Response
RESPONSE TO REVIEWER #1
Line numbers stated in the “Author Response” column corresponding to the manuscript file (File name: nutrients-1062334 v2), with the display under review set as ‘Simple Markup’. The amendments were indicated using red words below and with ‘track change’ in the manuscript file. Text background colour distinguishes between “author response” (with grey backgrounds) and “changes made to the text” (without grey backgrounds).
|
No. |
Reviewer#1 comment |
Author Response |
||||||||||||||||||||||||||||||||||||||||||||||||||||||||||||||||||||||||||||||||||||||
|
General comment |
Reviewer #1: · English language and style - Moderate English changes required. · Does the introduction provide sufficient background and include all relevant references? - Must be improved. · Is the research design appropriate - Yes. · Are the methods adequately described? - Yes. · Are the results clearly presented? – Can be improved. · Are the conclusions supported by the results? – Yes.
This study qualitatively assessed barriers and facilitators of mandatory nutrition labelling policy processes in Malaysia, and mapped the history of nutrition labelling initiatives internationally and in Malaysia. This is an important topic with implications for policy, and is appropriate for publication in Nutrients based on qualitative nutrition labeling analyses that have been published in the journal previously. I have included notes below on areas of the paper that I believe could be strengthened. |
Thank you for this comment. We have now improved the article based on your valuable comments and performed the necessary English editing.
|
||||||||||||||||||||||||||||||||||||||||||||||||||||||||||||||||||||||||||||||||||||||
|
1. |
ABSTRACT
Can you provide more context in the first sentence of the abstract explaining when mandatory nutrition labeling in Malaysia was implemented, and what “medium implementation” means? Could you also explain who conducted the recent food environment policy analysis? |
We have amended as per comment /Lines 15-31 (Abstract): Abstract: Mandatory nutrition labelling, introduced in Malaysia in 2003, received a ‘medium implementation’ rating from public health experts when previously benchmarked against international best practices by our group. The rating prompted this qualitative case study to explore barriers and facilitators during the policy process. Methods incorporated semi-structured interviews supplemented with cited documents and historical mapping of local and international directions up to 2017. Case participants held senior positions in the Federal government (n=6), food industry (n=3) and civil society representations (n=3). Historical mapping revealed that international directions stimulated policy processes in Malaysia but policy inertia caused implementation gaps. Barriers hindering policy processes included lack of resources, governance complexity, lack of monitoring, technical challenges, policy characteristics linked to costing, lack of sustained efforts in policy advocacy, implementer characteristics and/or industry resistance, including corporate political activities (e.g., lobbying, policy substitution). Facilitators to the policy processes were resource maximisation, leadership, stakeholder partnerships or support, policy windows and industry engagement or support. Progressing policy implementation required stronger leadership, resources, inter-ministerial coordination, advocacy partnerships and an accountability monitoring system. This study provides insights for national and global policy entrepreneurs when formulating strategies towards fostering healthy food environments. |
||||||||||||||||||||||||||||||||||||||||||||||||||||||||||||||||||||||||||||||||||||||
|
2. |
INTRODUCTION
p.2, lines 34-40: You say that population diets are “perceived as unhealthy”—who perceives them as unhealthy? Do you mean to say that those diets lead to NCDs?
|
We have amended Lines 35-41 (Introduction) to improve clarity as recommended and these changes are indicated below:
Unhealthy diets lead to overweight and obesity, and contribute to the development of non-communicable diseases (NCDs) [1-2], which were responsible for 11 million deaths and 255 million disability-adjusted life-years in 2017 [1]. In low- and middle-income countries (LMICs), 85% of premature deaths are from NCDs [3]. Population diets that contribute to the burden of obesity and NCDs, include suboptimal consumption of healthy foods (e.g., fruits and vegetables, nuts and seeds, milk and whole grains) and overconsumption of sugar-sweetened beverages, processed meat, red meat and ultra-processed foods containing high levels of negative nutrients (e.g., sodium, trans-fat) [1,4].
|
||||||||||||||||||||||||||||||||||||||||||||||||||||||||||||||||||||||||||||||||||||||
|
3. |
p.2, line 47: Can you provide an explanation of what you mean by “nutrition labelling” here? Is this a nutrition facts panel on the back of the package? Or front-of-package labeling? · Similarly, in the next paragraph, can you provide more explanation about what “nutrient declarations” are? |
We have amended Lines 57-75 (Introduction) to provide better clarity on the statements as below:
The World Health Organization (WHO) recommends implementation of mandatory laws and regulations to stipulate nutrition labelling on the back or front of food packages to align with the Codex Alimentarius Commission’s guidelines [8-9]. Previous reviews highlight that most consumers understand and use nutrition labelling during food purchase and selection [10-11], as well as recognise its role to support healthier diets [12-14]. A nutrition labelling policy governing pre-packaged foods is a cost-effective population intervention, enabling healthy food environments that encourage informed and healthier food choices [12,15]. In recent years, the World Trade Organization (WTO) has been urged to adopt WHO recommendations related to food labelling [2]. Disparities exist in how different countries approach the regulation of food labelling. Of the 124 WHO member countries, 85% have implemented nutrient declarations [16], requiring stating the nutritional content of a food product on the back of the package. However, in practice, policies vary across countries in the nutrients declared and food products for which labels must apply. In Malaysia, the Food Regulations 1985 mandates nutrient declarations only on frequently consumed packaged foods (e.g., bread, breakfast cereals, flour confection, canned products, fruit juices and soft drinks). In addition, foods with special purposes (e.g., infant formula), fortified foods and those carrying nutrition and health claims, as well as ready-to-drink beverages, are mandated to provide nutrient declarations [17]. The list of fully enforced mandatory declarations includes energy, carbohydrate, protein and fat content for frequently consumed packaged foods, together with total sugars content for ready-to-drink beverages and any claimed nutrients [17].
|
||||||||||||||||||||||||||||||||||||||||||||||||||||||||||||||||||||||||||||||||||||||
|
4. |
p.2, lines 55-62: Which nutrients are required to be listed under Malaysian regulations? |
We have amended Lines 70-75 (Introduction) as below:
… In addition, foods with special purposes (e.g., infant formula), fortified foods and those carrying nutrition and health claims, as well as ready-to-drink beverages, are mandated to provide nutrient declarations [17]. The list of fully enforced mandatory declarations includes energy, carbohydrate, protein and fat content for frequently consumed packaged foods, together with total sugars content for ready-to-drink beverages and any claimed nutrients [17].
|
||||||||||||||||||||||||||||||||||||||||||||||||||||||||||||||||||||||||||||||||||||||
|
5. |
p.2, line 70: What does “limited application” mean here? |
We have amended Lines 80-85 (Introduction) as below:
… The Food-EPI evaluation rated mandatory nutrition labelling with the highest degree of implementation (61%) but identified implementation gaps [20]. Implementation gaps included the lack of nutrient declarations for added sugar, sodium and saturated fat and that mandatory nutrition labelling only applies to frequently consumed packaged foods. The experts prioritised the need to expand the list of mandatory listed nutrients to include sodium and total sugars, as well as ‘added sugars’ [20].
|
||||||||||||||||||||||||||||||||||||||||||||||||||||||||||||||||||||||||||||||||||||||
|
6. |
p.2, lines 72-79: Did the INFORMAS report ultimately conclude that the policy was not properly implemented? Or is the takeaway that the policy itself should just be stronger? In other words, was the goal of your study to understand why the policy was not implemented completely, or why the policy didn’t include added sugar disclosure, etc? · Can you provide more context for why historical mapping of nutrition labelling policy was one of your goals? Has that not been documented previously? |
We have amended Lines 86-94 (Introduction) as below: The Food-EPI evaluation [18,20] highlighted that mandatory nutrition labelling policy in Malaysia was not optimally implemented, leaving room for improvement. This provided an opportunity to investigate “What are the enabling and limiting factors in the policy process?”, with the view to inform future efforts to shape and progress stronger food labelling policy. This case study aimed to (1) establish for the first time a historical mapping of nutrition labelling policy in parallel with cited international directions up to 2017 and (2) investigate barriers and facilitators in the policy process. Key lessons learnt from this case study on a South-East Asian and an upper-middle-income country will inform national, regional and global policy makers and related stakeholders about positioning strategies for healthy food environments.
|
||||||||||||||||||||||||||||||||||||||||||||||||||||||||||||||||||||||||||||||||||||||
|
7. |
METHODS
p.4, 2.1.3: If you interviewed 19-7 people (12), why were only 9 transcripts verified by interviewees? |
We have amended Lines 158-161 (Methods) to provide better clarity as below: 2.1.3. Stage 3: Data transcription, consolidation and analysis Audio-records were transcribed verbatim (SHN) and cross-checked by another researcher (TK) for logical consistency. Only nine participants elected to verify the transcripts, of which six participants made amendments after verification to improve clarity or censor statements to protect anonymity. |
||||||||||||||||||||||||||||||||||||||||||||||||||||||||||||||||||||||||||||||||||||||
|
8. |
RESULTS
p.4, 3.2. Part 1: What are “Big 4 declarations in 1985”? Can you provide more explanation about what Codex standards are? Are these general international actions on nutrition labeling unrelated to Malaysia, or are any of them specific to the country? |
We have amended Lines 192-210 (Results) as below, including the term of ‘Codex standards’ are now replaced by ‘Codex food label requirements’ to provide better clarity. 3.2. Part I: Historical mapping of mandatory nutrition labelling case Participants reported three key international directions occurring between 1985 and 1995. These included: (a) Codex Guidelines on Nutrition Labelling, promoting declarations on energy, carbohydrate, protein and fat content (termed as the ‘Big 4’) in 1985 [31]; (b) the World Declaration on Nutrition 1992 urging countries to harmonise with Codex food label requirements [32]; and (c) the establishment of the World Trade Organization (WTO) in 1995, which endorsed Codex guidelines [33]. Case participants noted a lack of significant response in Malaysia to these events during this period, although they did stimulate some policy discussion. Participants commented on the rising trends of overnutrition and NCDs from the 1980’s to 1990’s [34-36]. Some participants identified government-led prevention actions during this period, such as Healthy Lifestyle Awareness campaigns (1991-2002) that included education on reading food labels [37-38]. Some participants identified government actions during the period following the release of the international nutrition directives. They cited the National Plan of Action for Nutrition of Malaysia (NPANM) 1996-2000 which recommended to align with the Codex requirements for nutrition labelling [35]. In about the year 2000, the government proposed an amendment to the Food Regulations 1985, followed by multiple engagements with relevant stakeholders [39-42]. In 2003, mandatory nutrition labelling (P.U.(A)88, Reg.18B) was gazetted to extend the enforcement date to June 2005 [43-44]. The Guide to Nutrition Labelling and Claims was first published in 2005 [45] and updated twice [17,46] to facilitate policy implementation in Malaysia.
|
||||||||||||||||||||||||||||||||||||||||||||||||||||||||||||||||||||||||||||||||||||||
|
9. |
p.7, Table 1: Could you restructure Table 1 so that sub-themes are grouped by barriers and facilitators? |
We restructured Table 1 by first grouping barrier sub-themes within a theme together, followed by facilitator sub-themes (Lines 241-242, Results) as below: Table 1. Summary of barriers and facilitators as per stage of policy process.
Symbols: L = barrier; C = facilitator; √ = identified; X = not identified.
|
||||||||||||||||||||||||||||||||||||||||||||||||||||||||||||||||||||||||||||||||||||||
|
10. |
pp.7-11: It is difficult to read the direct text from all participants—could you summarize the subthemes instead of just listing all of the quotations, and only feature particularly salient quotations? |
We have amended Sections 3.3.1 to 3.3.7 (Lines 243-395) and featured only salient quotations. For quotations otherwise, important points were summarised in text without listing all the quotations. Lastly, we adjusted the text alignment of included quotations for clarity. 3.3.1. Policy commitment ‘Lack of resources’, the nature of ‘implementer characteristics’, and ‘lack of sustained effort’, were three barrier sub-themes that emerged in relation to policy commitment. ‘Resource availability or maximisation’, ‘supportive organisational action’ and ‘leadership’ were facilitator sub-themes. Participants reported ‘lack of resources’ such as technical knowledge and expertise, local evidence, guidelines, funding due to low prioritisation for monitoring and evaluation and/or laboratory capacity, which restricted policy processes. ‘Implementer characteristics’ linked to industries’ capacity and readiness to accept nutrition labelling hindered the policy process. Specific to policy implementation, competing priorities experienced by the enforcement team was another barrier linked to ‘implementer characteristic’. These issues were reflected in the following opinions: "… They [SME] do not understand [the new regulations] … [said] very difficult… [when] discussion, never came... … do not have QC [Quality Control] … … [and] wait until big companies to implement [first]...." (Civil society and industry, Development and Implementation). "… [based on] the hierarchy [of duties] … more enforcements related to the food safety… [rather than] nutrition labelling...", (Government, Implementation). A ‘lack of sustained effort’ by NGO advocacy during policy implementation was reported. Participants repeatedly highlighted concerned agencies lacked uniformity in messaging consumer education, as well as sustainability issues. In terms of facilitators, participants recognised that ‘resource availability or maximisation’ was critical throughout the policy processes. For instance, Codex guidelines, WHO recommendations and/or other ASEAN country experiences benefited stakeholders by facilitating the policy processes in Malaysia. ‘Supportive organisational action’ from top management and presence of structured systems modelled on the Codex Committees facilitated the policy process. Specific to policy development, ‘leadership’ from the government was cited as a facilitator sub-theme for mandatory nutrition labelling. A participant highlighted that: "… Last time, [nutrition labelling] was not mandatory… [but] Malaysia was the first country in ASEAN [Association of Southeast Asian Nations] to have mandatory .... We are bold enough to do that." (Civil society, Development). 3.3.2. Policy governance ‘Complexity’ and ‘lack of monitoring’ were two barrier sub-themes identified within policy governance specific to policy implementation. ‘Strategies in policy process’ was the only facilitator sub-theme identified within policy governance. ‘Complexity’ arises from issues in integrating labelling policy into existing regulatory frameworks that mainly mandated food safety. Participants cited complex bureaucratic procedures for legislation and competition with other policies as highlighted by this opinion: "I think there are internal issues, could be bureaucratic… political… legal. The Attorney General may come back and say, “You need to tackle [these], frame those words". Then, they have to take [the policy] back again and take actions." (Civil society, Implementation). Participants also acknowledged ‘lack of monitoring’ as another barrier during policy implementation. This was likely to be one of the contributing factors to policy inertia. ‘Strategies in the policy process’ was a facilitator sub-theme, which included media dissemination, development of guidebooks and permitting industry to negotiate for flexible grace periods for policy enforcement. Even though the Codex guidelines informed policy development and aligned with trading purposes, the Guidelines were adapted to the local context of Malaysia. Comments reflecting these views were: "It is not 100% we adopted the Codex Guidelines… Codex is the reference… [is useful] if there is a dispute at the WTO [World Trade Organization] … … we have [been] sensitising… we had the roadshows… discussion with the importers… called all the embassies to inform them." (Government, Development). “Two years grace period… the educational enforcement… in our guideline, even though you do not send for the lab analysis, you can use the food composition [database calculation method for labelling] … [and the guideline set] the analytical tolerance…” (Government and industry, Implementation). 3.3.3. External to policy organisation ‘Stakeholder partnership or support’ was the only facilitator sub-theme identified with this theme. This facilitator sub-theme was observed at the intra- and inter-ministerial levels (e.g., research institutions, trade and consumerism related agencies), as well as involving civil society members to advocate for the Big 4 declarations during the policy process. In addition, participants highlighted the role of the International Life Sciences Institute (ILSI), a non-profit organisation with members primarily from industry. A participant described ILSI’s role during policy process as: "ILSI decided, because this [Seminar] was regional... ILSI was willing to offer a platform, to bring in the various stakeholders…what's news in this area… … [bring] awareness and education to the professionals… [inviting] overseas speakers and [local government stakeholders].” (Government and industry, Development and Implementation). 3.3.4. Society Two sub-themes were found within the society theme. ‘Low demand or other attributes’ were identified as a barrier sub-theme, whilst the facilitator sub-theme was ‘social acceptance, awareness or benefit’. Participants frequently mentioned ‘low demand or other attributes’ related to consumer understanding and their limited use of nutrition labels, which they perceived to hinder policy processes. The following opinion reflected these views: "… the level of understanding in our consumers [was] different from the Western... ... at that time [of policy development] … [nutrition] labelling was very new... [and focused more on] underweight… … maybe they [consumers] read the label [now], but to what extent [do] they read the label?" (Civil society and government, Development and Implementation). In contrast, ‘social acceptance, awareness or benefit’ underpinned the development of nutrition labelling, particularly its positive effects linked to consumer education and healthy diets. 3.3.5. Industry Two opposing sub-themes were reported within the industry theme. ‘Industry resistance’ as the barrier sub-theme, whereas ‘industry engagement or support’ was the facilitator sub-theme. ‘Industry resistance’ was observed during the initial policy development step, contributing to limiting the focus to just the Big 4 declarations before broadening to include other nutrients of concern. Even after gazetting the mandatory nutrition labelling in March 2003, ‘industry resistance’ included the raising of issues such as waste from old packaging, which resulted in the extension of full enforcement of food regulations to June 2005. Nine non-industry participants remarked on the occurrence of industries’ corporate political activities. Strategies such as policy substitution and lobbying were cited by some participants to influence policy implementation, as reflected by the following views: Corporate political activity - policy substitution: "...related to nutrition declaration… they [i.e., industries] did not agree [with] the level… they proposed to have it in the Guideline, not in the regulations." (Government, Implementation). Corporate political activity – information and messaging (lobbying): "They [industries] are represented as a group... not by individual companies... Well, you can say they lobby...” (Civil society, Implementation). Participants with a civil society background expressed contrary views with regard to ‘lobbying’, with some identifying it as a positive contribution towards policy development, while others considered it as industry interference and manipulation of policy outcomes. These opposing views were reflected by the following comments: “… [as they are] speaking on front voices for the industry… we should not necessarily use the word "lobbying" in the negative sense…" versus “many lobbying… obviously not [only in] Malaysia but worldwide… powerful lobbying to make sure [policy] does not work… even developed countries… have to deal with this food lobbying, which is extremely powerful and very well coordinated.” Participants identified ‘industry engagement or support’ as a facilitator during the policy process because of the likely low negative impacts of limited nutrition labelling. Favourable opinions were the need for standardisation for a fair-trade environment, marketing opportunities with revenue benefits to industry on a long-term basis, availability of platforms for industry engagement and industry readiness factor in terms of ability to perform proximate analysis and/or have established products align to labelling standards. With regard to implementation and/or future plans, industry participants in this study showed positive views on the mandatory nutrient list expansion. Possible reasons cited were: "Because most of our customers [e.g., foreign retailers] required us to provide based on the importing country’s regulation, [which the] requirement is higher [than] the Malaysian [standards]." (Industry, Implementation and/or future plan). "[Nutrition labelling] is part of the cost of goods, but it is one-off… It is not very expensive to analyse sodium and total sugars [contents]… Printing is routine… If [the] right transition period [is provided], there is not really a big issue because most of the companies are changing their packaging." (Industry, Implementation and/or future plan). 3.3.6. Policy specific issue Under the policy specific issue theme, there were two sub-themes specific to barriers. These included ‘policy characteristics’ and ‘technical challenges’. No facilitator sub-theme was identified. ‘Policy characteristics’ linked to costing (e.g., analytical, labour and printing costs), uncertainties in specifications like labelling format and implementation timeline, as well as slow regulatory amendment procedures which hindered the policy process. Comments reflecting these views were: "…. to [standardise the] label, of course there will be some costing. Because [industries] have to send to lab for analysis… change labels… " (Government, Development). "… the analytical declaration tolerance, it [was] only after implementation, we [industry] realised it [as a problem]. There was quite a bit of discussion." (Industry, Implementation). "… the amendment was quite slow… we pushed for sodium and also sugars [labelling] for quite some time." (Government, Implementation and/or future plan). ‘Technical challenges’ hindered the policy processes. These included a lack of laboratory capacity, analytical limitations, a non-comprehensive food composition database for nutrients of concern, stock turnover issues and/or challenges related to standards’ harmonisation. 3.3.7. Opportunistic advantages ‘Policy windows’ and ‘revenue-related effects’ emerged as two sub-themes for opportunistic advantages, both of which were discussed as facilitators. Rising obesity and NCD rates triggered the policy process. The international directions (e.g., Codex or WHO related opportunities), coupled with local events such as unregulated claims, the appointment of Food Safety Quality Division as the Codex contact point and lobbying by international academia for salt reformulation contributed to facilitating the policy process. All of these factors formed ‘policy windows’ for mandatory nutrition labelling in Malaysia. ‘Revenue related effects’ facilitated the policy development of mandatory nutrition labelling. Participants’ comments were influenced by their backgrounds, such as: “Driving force... Of course, it is facilitating the trade as well." (Government, Development). “… companies feel… "If I do not have it, I lose out to the other companies." (Civil society, Development).
|
||||||||||||||||||||||||||||||||||||||||||||||||||||||||||||||||||||||||||||||||||||||
|
11. |
DISCUSSION
Much of the discussion summarizing the overall results could be moved to the results section (see my previous comment), and then the discussion could focus more on interpretation and comparing results to other studies. |
We have amended the ‘Discussion’ section in response to this comment / Lines 426-440, accordingly as below: The government’s early development of nutrition labelling policy clearly reflected and followed the international directives. Malaysia fulfilled the international commitments [32-33] and pioneered mandatory nutrition labelling amongst the South-east Asian countries, harmonising this with Codex Guidelines on Nutrition Labelling 1985 [65-66]. The enactment of the Big 4 declarations in 2003 was the initial step to strengthen nutrition labelling in the marketplace. In contrast, more developed countries such as Australia and New Zealand mandated additional nutrient declarations (i.e., sodium, sugars and saturated fat) as part of their public health prevention efforts [67] during the same period. The question is - Why was nutrition labelling only limited to the Big 4 declarations? This may reflect local constraints in Malaysia but could also be attributed to Codex guidelines setting minimum requirements to enable less-developed nations achieve primary food safety as part of provisions to facilitate trade [68]. However, the limited nutrient declarations may also reflect constraints from political alliances and the presence of the food industry front group represented by ILSI during Codex meetings, as has been reported previously [68]. Thow et al. [69] shared concerns on the high representation of food industry compared to public health advocates in Codex meetings and called for the latter to raise awareness of industry influence over domestic policy makers.
|
||||||||||||||||||||||||||||||||||||||||||||||||||||||||||||||||||||||||||||||||||||||

Reviewer 2 Report
Referee Report on Manuscripts ID: Nutrients-1062334
“Tracking progress from policy development to implementation: A case study on adoption of mandatory regulation for nutrition labelling in Malaysia”
The paper is, generally, well-written and properly structured. But I do have some comments, which I outline below:
1. The introduction section needs to be improved. Here are some recommendations:
A. Although this study emphasizes mandatory food labeling, the introduction section does not provide information about labeling. What is labeling? Why do we need labeling? What are the advantages of food labeling? All these discussions may create enough background for readers to understand the importance of adopting mandatory labeling.
Check the following paper:
Bonroy O. and C. Constantatos. 2014. “On the Economics of Labels: How Their Introduction Affects the Functioning of Market and the Welfare of All Participants.” American Journal of Agricultural Economics, 97(1): 239-259.
B. Are there any disadvantages of introducing a labeling policy? The introduction of labeling policy resolves the asymmetric information problem. However, the introduction of labeling may create incentives for fraudulent behavior of producers in the form of mislabeling / misrepresentation because of imperfect monitoring and enforcement system. For the completeness of the discussion, it is important to mention the caveats of introducing labeling. The following paper shows the caveats related to labeling policy.
Meerza, S.I.A.; Giannakas, K.; Yiannaka, A. Markets and welfare effects of food fraud. Aust. J. Agric. Res. Econ. 2019, 63, 759–789.
2. Did you pay participants? What was the average length of the face-to-face interview?
3. Section 3.3.8 requires discussion.
4. The conclusion section may include future research directions.
I hope you find my comments helpful in revising this research.
Author Response
RESPONSE TO REVIEWER #2
Line numbers stated in the “Author Response” column corresponding to the manuscript file (File name: nutrients-1062334 v2), with the display under review set as ‘Simple Markup’. The amendments were indicated using red words below and with ‘track change’ in the manuscript file. Text background colour distinguishes between “author response” (with grey backgrounds) and “changes made to the text” (without grey backgrounds).
|
No. |
Reviewer #2 comment |
Author Response |
|
General comment |
Reviewer #2: · English language and style – English language and style are fine/ minor spell check required. · Does the introduction provide sufficient background and include all relevant references? - Must be improved. · Is the research design appropriate - Yes. · Are the methods adequately described? – Can be improved. · Are the results clearly presented? – Yes. · Are the conclusions supported by the results? – Yes.
“Tracking progress from policy development to implementation: A case study on adoption of mandatory regulation for nutrition labelling in Malaysia”. The paper is, generally, well-written and properly structured. But I do have some comments, which I outline below. |
Thank you for the comments. We improved the article according to your useful comments and provided English editing.
|
|
1. |
Introduction
The introduction section needs to be improved. Here are some recommendations: A. Although this study emphasizes mandatory food labeling, the introduction section does not provide information about labeling. What is labeling? Why do we need labeling? What are the advantages of food labeling? All these discussions may create enough background for readers to understand the importance of adopting mandatory labeling.
Check the following paper: Bonroy O. and C. Constantatos. 2014. “On the Economics of Labels: How Their Introduction Affects the Functioning of Market and the Welfare of All Participants.” American Journal of Agricultural Economics, 97(1): 239-259.
B. Are there any disadvantages of introducing a labeling policy? The introduction of labeling policy resolves the asymmetric information problem. However, the introduction of labeling may create incentives for fraudulent behavior of producers in the form of mislabeling / misrepresentation because of imperfect monitoring and enforcement system. For the completeness of the discussion, it is important to mention the caveats of introducing labeling. The following paper shows the caveats related to labeling policy.
Meerza, S.I.A.; Giannakas, K.; Yiannaka, A. Markets and welfare effects of food fraud. Aust. J. Agric. Res. Econ. 2019, 63, 759–789. |
We improved the Introduction as recommended /Lines 48-64 (Introduction):
Labelling is a policy instrument to guide the food industry to present specific-food product information related to country of origin, environmental protection and consumer health [6]. Some label criteria distinguish high-quality products from uninspected or unaccredited products [6], allowing consumers to make product comparisons. Logically, food labelling policy sets the standard requirements for both locally manufactured and imported products in the market, minimising issues related to asymmetric product information. However, a theoretical framework analysis suggests that even though a policy may exist, fraudulent behaviour of producers will still be observed as regards improper product information [7]. The existence of such fraudulent behaviour likely depends on the enforcement system, public complaints on label violations and economic drivers for food fraud [7]. The World Health Organization (WHO) recommends implementation of mandatory laws and regulations to stipulate nutrition labelling on the back or front of food packages to align with the Codex Alimentarius Commission’s guidelines [8-9]. Previous reviews highlight that most consumers understand and use nutrition labelling during food purchase and selection [10-11], as well as recognise its role to support healthier diets [12-14]. A nutrition labelling policy governing pre-packaged foods is a cost-effective population intervention, enabling healthy food environments that encourage informed and healthier food choices [12,15]. In recent years, the World Trade Organization (WTO) has been urged to adopt WHO recommendations related to food labelling [2].
References: [6] Bonroy, O.; Constantatos, C. On the Economics of Labels: How Their Introduction Affects the Functioning of Market and the Welfare of All Participants. Am J Agric Econ. 2014, 97(1), 239-259.
[7] Meerza, S.I.A.; Giannakas, K.; Yiannaka, A. Markets and welfare effects of food fraud. Aust J Agric Resour Econ. 2019, 63, 759–789.
|
|
2. |
Did you pay participants? What was the average length of the face-to-face interview? |
We have amended in response to the comment /Lines 156-157 (Materials and Methods):
All interviews were conducted in English and audio-recorded with written notes to support the recording. The interview was voluntary and did not involve any monetary payment to participants.
For the average length of interview, we already indicated this information in Line 192-193 (Results).
… Interviews lasted an average length of 1 hour 14 minutes. |
|
3. |
Section 3.3.8 requires discussion. |
We provided a description of recommendations in Lines 396-401 (Results): 3.3.8. Recommendations The interviews also generated recommendations from participants to facilitate future food labelling policy processes (Table 2). Actions recommended by participants included to enhance consumer education, determine an appropriate transition period for full enforcement, maximise resources particularly scientific evidence, publish clear guidelines with stakeholders’ engagement and proper training, and to intensify accountability systems.
We improved Lines 480-507 (Discussion) to strengthen the discussion related to Section 3.3.8 (Recommendation):
Seeking participants’ recommendations based on their policy experiences provides valuable guidance for future food policy actions. Participants emphasised the need for independent monitoring as well as maximising resources, especially credible evidence (e.g., Codex guidelines, WHO recommendations and local research) to progress future food labelling policy processes. Literature from the same region [72,73] supports similar views on the importance of resources to inform policy decisions, alongside a monitoring and evaluation mechanism to identify the gaps in food environment policy implementation. In addition, a majority of the participants recommended strengthening consumer education activities in promoting awareness and use of nutrition labelling. A systematic review of nutrition label education studies in Western countries supports this recommendation and found that consumer education could positively impact a consumer’s label understanding and its use [82]. Over time, these impacts are anticipated to establish impetus for the government and food industry to change the status quo, fulfilling the social and market demands for comprehensive nutrition labelling. Implementation of the mandatory nutrition labelling policy still warrants further improvement in Malaysia, despite progressive policy steps over the last few decades. This was in spite of past experiences related to the labelling area, increasing market demands for comprehensive labelling and establishing optimal laboratory capacity, either in-house or outsourced to accredited laboratories. For instance, nearly half of prominent food companies (13 out of 28) had declared total sugars content on some products [75], well before full enforcement of the new regulation amendments in 2022 [70]. For food industries yet to implement the policy, participants recommended that stock turnover issues could be tackled with an appropriate grace period. Drawing from past Malaysian experience, a maximum of a 2-year transition period to enforce total sugars and sodium declarations as per the recent gazette [70] would enable businesses to phase out existing label inventory and align newer labelling with mandatory requirements. Such a grace period duration is in line with the recommendation of the Codex Committee on Food Labelling [63] for governments. Participants also recommended the need for clear guidelines and industry engagement and training, particularly to support SME businesses. This study’s recommendations are in line with the Codex views [63] towards creating effective communication strategies to allow synergistic effects for progressing policy implementation. |
|
4. |
The conclusion section may include future research directions.
I hope you find my comments helpful in revising this research. |
Thank you for the valuable comments and we have amended accordingly in Lines 526-538 (Conclusions):
5. Conclusions This study adds insights into the barriers and facilitators in the mandatory nutrition labelling policy processes from a LMIC perspective. The case study revealed the main influences on the policy processes to be policy commitment, governance and its technical and specificity issues, stakeholders’ relationships, social attributes and impacts, food industry’s policy position, as well as opportunities linked to local and external triggers influenced policy processes. Policy inertia was evident in this Malaysian experience. Key lessons gained from this study can inform policy entrepreneurs, particularly in LMICs, to understand considerations of adopting mandatory nutrition labelling and formulate strategies to mitigate challenges and seize opportunities to create healthy food environments. Future research directions are necessary to examine the impact of nutrition labelling policies on reformulation, sales and consumer behaviours using quantitative analyses, evaluate corporate political activities of food companies and peak bodies, as well as related influences on policy inertia and mechanisms to manage conflicts of interest.
|

Round 2
Reviewer 2 Report
Thank you for revising the paper.